# Monitoring Immune Modulation in Season Population: Identifying Effects and Markers Related to *Apis mellifera ligustica* Honey Bee Health

**DOI:** 10.3390/biom14010019

**Published:** 2023-12-22

**Authors:** Olga Frunze, Hyunjee Kim, Byung-ju Kim, Jeong-Hyeon Lee, Mustafa Bilal, Hyung-Wook Kwon

**Affiliations:** 1Department of Life Sciences, Incheon National University, 119 Academy-ro, Yeonsu-gu, Incheon 22012, Republic of Korea; frunzeon@gmail.com (O.F.); beamed79@hanmail.net (H.K.); bjk@inu.ac.kr (B.-j.K.); jhl2532@naver.com (J.-H.L.);; 2Convergence Research Center for Insect Vectors (CRCIV), Incheon National University, 119 Academy-ro, Yeonsu-gu, Incheon 22012, Republic of Korea

**Keywords:** honey bee, health markers, innate immunity, defense system, antioxidant system, ROS enzymes, *Apis mellifera ligustica*

## Abstract

Honey bees play a significant role in ecology, producing biologically active substances used to promote human health. However, unlike humans, the molecular markers indicating honey bee health remain unknown. Unfortunately, numerous reports of honey bee collapse have been documented. To identify health markers, we analyzed ten defense system genes in *Apis mellifera ligustica* honey bees from winter (Owb) and spring (Fb for foragers and Nb for newly emerged) populations sampled in February and late April 2023, respectively. We focused on colonies free from *SBV* and *DWV* viruses. Molecular profiling revealed five molecular markers of honey bee health. Of these, two seasonal molecular markers—*domeless* and *spz* genes—were significantly downregulated in Owb compared to Nb and Fb honey bees. One task-related marker gene, *apid-1*, was identified as being downregulated in Owb and Nb compared to Fb honey bees. Two recommended general health markers, *SOD* and *defensin-2*, were upregulated in honey bees. These markers require further testing across various honey bee subspecies in different climatic regions. They can diagnose bee health without colony intervention, especially during low-temperature months like winter. Beekeepers can use this information to make timely adjustments to nutrients or heating to prevent seasonal losses.

## 1. Introduction

Honey bees play a significant role in ecology and produce biologically active substances that humans have long used to promote health. Various aspects of health, including at the molecular, microbial, and population levels, have recently gained increased interest from researchers, stakeholders, and citizens [1]. However, unlike humans, the standard parameters of honey bee health remain unclear. To investigate this, it is crucial to consider the different castes and ages of honey bees present in a colony, including the queen, drone, nurses, and forager workers [2,3], as well as the two types of workers: the long-lived winter population and the short-lived summer population [4]. While short-lived honey bees live for about 35–45 days, long-lived honey bees can survive up to six months [5]. Short-lived workers perform different tasks in an age-dependent sequence: newly emerged bees transform into young workers (nurses), which usually care for the brood during the first two or three weeks of adult life, while older workers (foragers) forage for nectar and pollen outside the nest [6]. In contrast to short-lived bees, long-lived honey bees form winter clusters, enabling colony thermoregulation during the 3–4 months of winter. After overwintering, they begin to exhibit a division of labor similar to that of short-lived bees, contributing to the rearing of the new generation of worker bees [7]. This full exchange of winter bees to spring bees lasts until mid-April in the Republic of Korea.

It is no wonder that winter and spring *Apis mellifera* honey bees exhibit different immune responses to virus infections [4]. Insects, in general, have a well-developed innate immune system comprising humoral and cellular responses [8]. Humoral defenses refer to soluble effector molecules such as antimicrobial peptides, complement-like proteins, and enzymatic cascades that regulate melanin formation and clotting [9] through the four well-known signal transduction pathways of innate immunity: Toll, Imd, JNK, and JAK/STAT [8,10]. They are related to reactive oxygen species (ROS) enzymes, which reduce oxidative stress through antioxidative defense mechanisms [11]. A previous research group reported that the season was the main factor responsible for changes in antioxidant enzymes [11]. Steinmann et al. concluded that the physiological adaptation of the winter population of honey bees to increase survival is associated with an overall decrease in physiological activity [5]. However, this downregulation of the energetically expensive immune system results in increased susceptibility to pathogens. This phenomenon is evidently related to reports about honey bee collapse, where the most significant colony losses occur during overwintering [1]. Importantly, both generations of honey bees share the same genetic background, and differences in health and immunity develop only due to the influence of the annual seasonal cycle. It is crucial to note that the immune response of honey bees depends on different tasks and season-linked variability, although this mechanism has not yet been fully clarified.

In honey bees and other organisms, specific patterns at the gene and gene transcript levels [12] serve as biomarkers [13] and are termed “molecular profiling”. While each biomarker reflects the cumulative response of complex molecular interactions, it can be detected through correlation analysis using reporter genes for pathway analysis [14]. In this research, the molecular monitoring of honey bee health was conducted for the first time through the quantification and correlation (molecular profiling) of selected reporter genes from defense system pathways. The health of both long-lived (winter population) and short-lived (spring population) honey bees was studied. Furthermore, among the short-lived honey bees, we examined task-specific differences in newly emerged (Nb) and forager (Fb) healthy honey bees. This comparison was made because the physiology of overwintered bees (Owb) was assumed to be similar to the task and gland development of newly emerged bees, despite the chronological age of Owb being greater than that of Fb. We analyzed 10 genes from the defense system: *spz* and *dorsal-1* (Toll pathway), *PGRP-LC* and *relish* (Imd pathway), *domeless* (JAK/STAT pathway), *defensin-2* and *apid-1* (antimicrobial peptides, AMP), *SOD*, *SOD2*, and *Trxr1* (antioxidative defense genes coding ROS enzymes). We identified two genes as seasonal health markers, one gene as a task-related marker, and two genes as general health markers. Furthermore, we found that the correlation between ROS enzyme genes and innate immunity genes was related to task specificity. This research highlights the defense response mechanism, paving the way for the development of a molecular system to screen honey bee health. It focuses on understanding the longevity of long-living bees compared to short-lived ones and aims to protect them from losses in the future.

## 2. Materials and Methods

### 2.1. Experimental Bees

Honey bees (*Apis mellifera ligustica*) were obtained from twelve colonies kept at the Gangneung apiary in the Republic of Korea. Experimental colonies were managed by a professional beekeeper following standard beekeeping practices. Samples for analysis were collected from selected colonies that did not exhibit clinical symptoms of diseases or infestation by *Varroa destructor*. The winter honey bee population was sampled in the middle of February 2023 when queens had not yet started laying eggs, and the overwintered bees were expected to have successfully survived the winter and were about to start feeding the first larvae. Bees from the spring population (newly emerged and foragers) were collected in late April 2023 from the same colonies; this period is considered to be the two weeks before the peak of the beekeeping season.

### 2.2. RNA Extraction and cDNA Synthesis

A total of 90 honey bees (30 winter bees, 30 newly emerged bees, and 30 foragers with pollen) from each *Apis mellifera ligustica* colony were sampled and stored at −80 °C. Three honey bees from each colony were analyzed. They were added to CKmix50_2 mL tubes (Precellys Lysing Kit) containing ceramic beads (MP Biochemicals GmbH, Eschwege, Germany). Cold RLD buffer (1 mL) was added to the tubes, and they were homogenized using a Precellys Evolution Tissue Homogenizer (Bertin Instruments, Montigny-le-Bretonneux, France) following the manufacturer’s recommended program. A total of 600 µL supernatant with debris was transferred to 1.5 mL Eppendorf tubes, and 400 µL of cold RLD buffer was added. The subsequent steps of RNA isolation followed the Quick-Start Protocol (Clear-STM Total RNA Extraction Kit, in VIRUStech, Republic of Korea). The final volume was 30 µL of total RNA in Elution Buffer. The total RNA concentration and purity were quantified using OD260/OD280 values between 1.8 and 2.0. Next, reverse transcription was performed using the RNA to cDNA EcoDryTM Premix (Oligo dT) kit (Takara, San Jose, CA, USA). Each reverse transcription reaction mixture included 50 ng/µL of total RNA (with the clear volume calculated for each sample) and RNase-free water for a total volume of 20 µL. Reverse transcription was conducted at 42 °C for 60 min, followed by heating at 70 °C for 10 min.

### 2.3. Quantitative Real-Time PCR

Relative expression was measured for ten genes of the defense system: *spz*, *dorsal-1*, *PGRP-LC*, *relish*, *domeless*, *defensin-2*, *apid-1*, *SOD*, *SOD2*, and *Trxr1*. Gene for *β-actin* was used as an endogenous control. PCR primer sequences are shown in Table 1.

The reaction conditions were optimized. The qRT-PCR reaction volume of 20 µL included 2 µL of template cDNA, 10 µL of 2× GreenStar Master Mix, 1 µL of upstream and downstream primers (5 pM/µL), and 6 µL of ddH_2_O. Quantitative real-time PCR (qRT-PCR) was conducted using an AccuPower 2× GreenStarTM qPCR Master Mix (BIONEER, Oakland, CA, USA) on a AriaMx Real-Time PCR System (Agilent Technologies LDA, Bayan Lepas, Malaysia).

The qRT-PCR amplification conditions were as follows: an initial denaturation at 95 °C for 10 min, followed by 40 cycles of denaturation at 95 °C for 30 s, annealing at 60 °C for 25 s, and extension at 72 °C for 15 s. Each sample was technically replicated three times. Data analysis was performed using Agilent AriaMx version 2.0 Analysis Software. Relative gene expression data were analyzed using the 2^(−Delta Delta C(T)) method [15]. Confirmation of *β-actin* and amplicons in the RT-PCR products was performed by separating them through electrophoresis in a 1% agarose gel at 100 V for 20 min and analyzing them using a gel documentation system, the Gerix 1010 transilluminator (Biostep GmbH, Burkhardtsdorf, Germany). A 100 bp DNA Ladder (BIONEER, BIO-RAD, Seoul, Republic of Korea) was used as a reference.

### 2.4. Molecular Detection of SBV and DWV to Exclude Data from Diseased Honey Bees

We detected the presence of Sacbrood Virus (SBV) and Deformed Wing Virus (DWV) diseases using the qPCR (primers presented in Table 1) and the data of honey bees with positive infection were deleted. So, we expect that the data used for analysis belong to relatively healthy honey bees.

### 2.5. Statistical Analysis

The statistical analysis, designed as illustrated in Figure 1, was conducted using Microsoft Excel 2016 and XLSTAT software (Addinsoft Pearson Edition 2014, Addinsoft, Paris, France). Specifically, the categorization of honey bees into groups was plotted using Discriminant Analysis (DA). Although DA is one of the most common data reduction techniques, it suffers from two main problems: Small Sample Size (SSS) and linearity problems [16]. Therefore, DA needs to be complemented by other statistical analyses [17]. The mean, standard deviation, and variance of each gene were calculated using descriptive statistics and visualized using the Heat map module. Gene expression variation was measured as the % coefficient of variation (CV) of gene expression [18]. For the gene expression analysis, ANOVA was used to test overall effects, followed by the Duncan post hoc test (*p* < 0.05) for multiple comparisons of gene expression between groups of honey bees.

The bar plots were generated using the online SRplot Official Website https://www.bioinformatics.com.cn/en (accessed on 30 October 2023). Canonical relationships among the ten traits were plotted to illustrate group differences. Pair-wise correlation analysis was conducted using the Pearson method [19] to identify significant relationships between the traits (*p* < 0.05). The correlations were interpreted based on the guidelines provided by Hinkle et al., categorizing correlations as very high positive (negative) correlation (±0.90 to 1.00), high positive (negative) correlation (±0.70 to 0.90), moderately positive (negative) correlation (±0.50 to 0.70), low positive (negative) correlation (±0.30 to 0.50), and negligible correlation (±0.00 to 0.30) [19].

## 3. Results

### 3.1. Molecular Profiling with Defense Gene Expressions

Molecular profiling of the defense system of honey bees that were sampled in February (Owb, overwintered honey bees, winter population) and in April (newly emerged bee, Nb; forager bee, Fb; spring population), was performed by analyzing the expression of ten genes: *spz* and *dorsal-1* (Toll pathway), *PGRP-LC* and *relish* (Imd pathway), *domeless* (JAK/STAT pathway), *defensin-2* and *apid-1* (antimicrobial peptides, AMPs), *SOD*, *SOD2*, and *Trxr1* (antioxidative defense genes coding ROS enzymes).

#### 3.1.1. Selection of the Markers Based on Discriminant Analysis Scores

The Owb, Nb, and Fb honey bees were identified based on comparisons of the 10 gene expressions (Figure 2) via Canonical Discriminant Analysis (DA). Significant Pillai’s trace criterion (Value: 1.293; df1: 18; df2: 46; *p* < 0.0001) determined that DA was feasible. As reported in Appendix A, out of the ten discriminant functions designed after the analysis, five presented a significant discriminant ability (*spz*, *domeless*, *Trxr1*, *apid-1*, and *PGRP-LC*). The discriminatory power of the F1 function was high (eigenvalue of 5.647), with 87.65% of the total variation, and 12.35% of the remaining variation achieved by the second function (F2) (Figure 2). The confusion matrix for the cross-validation results revealed a clear definition of Owb honey bee samples, but only 66.67% of Fb and 53.33% of Nb honey bees were placed correctly (Appendix A).

Variables were ranked depending on their discriminating properties. For this, classification functions (the linear discriminant function) across each group were used because they are helpful in deciding which variable affects classification more. Out of the ten significantly discriminant functions, only the two (*spz* and *domeless*) most relevant functions and, hence, marker candidates, were used to build a standardized discriminant coefficient biplot to differentiate the Owb, Nb, and Fb honey bees.

#### 3.1.2. Effect of Variation on Marker Selection

Genes with stable expressions were specified via coefficient of variation (CV) analysis. CV below 1.0 (bolded type in Table 2) was accepted for selection of the general genes *spz*, *domeless*, *apid-1*, *relish*, and *SOD* as marker candidates.

#### 3.1.3. Selection of the Markers Based on Gene Expression

A one-way ANOVA and Duncan post hoc tests were performed to compare the effect of population (winter vs. spring) and task specificity (newly emerged vs. forager) on the defense systems of honey bees via ten gene expressions. It was revealed that immune gene expressions of the Toll pathway (*spz* and *dorsal-1*) led to a significant difference in *spz* between winter (downregulated) and spring (upregulated) honey bee populations (F(2, 28) = 4.459, *p* = 0.02 (*p* < 0.05)). The immune gene expression of the JAK/STAT pathway caused a significant difference in *domeless* between winter (downregulated) and spring (upregulated) honey bee populations (F(2, 28) = 39.230, *p* < 0.0001 (*p* < 0.05)). Additionally, the antimicrobial peptide (AMP) *apidaecin-1* (*apid-1*) exhibited significant differences. *Apid-1* expression was downregulated in Nb and Owb, whereas it was upregulated (F(2, 28) = 6.080, *p* = 0.006, *p* < 0.05) in Fb honey bees. There was no statistically significant difference in ROS enzyme (*SOD*, *SOD2*, and *Trxr1*) gene expressions between honey bees (*p* > 0.05) (Figure 3).

The highest predominant transcripts of the gene expression value were ROS enzyme *(SOD*) and AMP gene (*defensin-2*). Both of them were not dependent on honey bee generations and their expressions dominated over the other eight genes.

Next molecular markers of honey bee health were selected based on three criteria: high DA score, CV lower than 1, and significant differences in gene expression (*p* < 0.05). The molecular markers of health were identified as *spz*, *domeless*, and *apid-1* (Figure 4). The molecular profile of Owb (winter population, overwintered) depicted the diapause condition of honey bees, where the recognition patterns of Toll (*spz*), JAK/STAT (*domeless*) pathways, and AMP protein (*apid-1*) genes were significantly downregulated. This implies a special defense mechanism related to the longevity of the winter population of honey bees. The molecular profile of Nb and Fb (spring population, newly emerged, and forager) honey bees showed significantly upregulated genes in the recognition pattern of Toll (*spz*) and JAK/STAT (*domeless*) pathways. However, AMP protein (*apid-1*) was downregulated in Nb, but upregulated in Fb honey bees.

So, molecular profiling revealed five molecular markers of honey bee health. The two seasonal molecular markers were *domeless* and *spz* genes with significant downregulation in Owb; one task-related marker gene, *apid-1*, was upregulated in Fb; and two general health markers, *SOD* and *defensin-2*, were upregulated in honey bees. These markers need further testing across various honey bee subspecies in different climatic regions.

### 3.2. Molecular Profiling of Defense System Based on Correlation Analysis

The objective of the correlation network analysis was the monitoring of honey bee health through the selected pathway reporter genes (Appendix A). We observed how the interaction between the immune response genes (ROS enzyme and innate immune gene expressions) was altered in relation to populations (winter and spring) and tasks (newly emerged and foragers).

The honey bee winter population Owb had 13 total significant moderate and higher correlations. It was fewer than the total number of correlations of Nb (18) and more than the total number of correlations of Fb (8) (Figure 5).

#### 3.2.1. The Common Correlations between Defense Genes in Honey Bees

We detected five common correlations of Owb with Nb honey bees and one of Owb with Nb honey bees, suggesting a greater similarity of Owb molecular interactions to Nb than to Fb (Figure 5). Comparing the Owb and Nb honey bees, the three common correlations of *spz* with *domeless* (*p* = 0.922, 0.521, *p* < 0.05) and with *PGRP-LC* (*p* = 0.818, 0.687, *p* < 0.05) and *dorsal-1* with *SOD2* (*p* = 0.677, 0.996, *p* < 0.05) were positive, and one common correlation between *spz* with *defensin-2* was negative (*p* = −0.930, −0.687, *p* < 0.05), respectively. However, a fifth correlation between *dorsal-1* and *apid-1* was positive in Owb (*p* = 0.719, *p* < 0.05) and negative in Nb (*p* = −0.694, *p* < 0.05). Interestingly, in Owb and Nb honey bees, all five common correlations included one gene from the Toll pathway (*spz* or *dorsal-1*), which indicates the high regulatory activity of this pathway in the immune response mechanism of both honey bee populations (Figure 6A,C).

Additionally, in Owb compared to Fb, the common correlation between *defensin-2* and *apid-1* (*p* = −0.852; *p* < 0.05) was negative in the first, opposite to the positive correlation (*p* = 0.856; *p* < 0.05) observed in the second honey bee group. Also, we observed a decrease in the number of correlations from 18 to 8 after approximately 14 days of development from Nb to Fb honey bees. However, compared to Nb with Fb, four common correlations were identified (Figure 6C,E). Two positive correlations were between *relish* with *dorsal-1* (*p* = 0.985, 0.847, *p* < 0.05) and with *SOD2* (*p* = 0.984, 0.702, *p* < 0.05) in Nb and Fb, respectively. Moreover, the two correlations between *domeless* with *relish* and with *dorsal-1* were negative (*p* = −0.730, *p* < 0.05) and (*p* = −0.740, *p* < 0.05) in Nb, but they were positive (*p* = 0.849, *p* < 0.05) and (*p* = 0.894, *p* < 0.05) in Fb honey bees. We suggest that reversing the correlation might be task-dependent in the immune response mechanism.

#### 3.2.2. The Specific Correlations between Defense Genes in Honey Bees

There were seven specific significant correlations in Owb, with four of them being positive and three being negative (Figure 6B). The correlation coefficients for *spz* with *apid-1*, *domeless* with *PGRP-LC* and with *apid-1*, *defensin-2* with *PGRP-LC* and with *domeless*, and *PGRP-LC* with *relish* and with *apid-1* were 0.968, 0.762, 0.868, −0.817, −0.840, −0.691, and 0.702 (*p* < 0.05), respectively. Nine specific correlations were identified in Nb. The correlation coefficients between *Trxr1* with *dorsal-1*, *relish*, *SOD2*, *domeless*, and *apid-1*, as well as *SOD2* with *domeless* and *apid-1*, and *SOD* with *domeless* were 0.901, 0.814, 0.888, −0.667, −0.600, −0.728, −0.691, and −0.573 (*p* < 0.05), respectively. Notably, every correlation in Nb included one gene from the group of ROS enzymes (Figure 6D and Appendix A).

Three specific negative correlation coefficients were detected in Fb (Figure 6F): between *Trxr1* with *defensin-2* and *SOD*, and *SOD* with *spz* were −0.744, −0.679, and −0.945 (*p* < 0.05), respectively. Interestingly, each correlation in Nb and Fb honey bees had a relationship to ROS enzymes, indicating the active regulation between ROS enzymes with innate immunity pathways in the honey bee spring population.

The healthy Owb (winter population) honey bees were characterized by thirteen significant moderate-to-higher correlations among ten defense system genes. Specifically, 54% of these interactions pertained to innate immunity genes, which evidently influence the longevity mechanisms of the fall generation of honey bees (Figure 6B). Key genes in this mechanism included *PGRP-LC* (Imd pathway), *domeless* (JAK/STAT pathway), *apid-1*, and *defensin-2* (both antimicrobial peptides), each showing multiple correlations. Moreover, Owb had five common correlations with Nb (Figure 6C) compared to one with Fb honey bees. This might indicate that due to their enduring youthful physiology, Owb bees (long-lived), despite being overwintered, had not yet raised a brood and maintained a youthful physiology akin to newly emerged bees (Nb).

The healthy Nb (spring population) revealed eighteen significant correlations among ten defense genes. Over 50% involved ROS enzyme genes (*SOD*, *SOD2*, and *Trxr1*), crucial for Nb’s pathogen response. Notably, their negative correlations with the recognition of the *domeless* (JAK/STAT pathway) and *apid-1* (AMP) genes indicated a protective mechanism from overexpression in the case of high stimulation. Four of the same correlations in Nb and in Fb were observed. Two of them remained positive: *relish* (Imd pathway) with *SOD2* and with *dorsal-1* (Toll pathway). However, correlations of *domeless* (JAK/STAT pathway) with *relish* and with *dorsal-1* were reversed from negative in Nb to positive in Fb. This molecular profiling highlighted the task-specificity of the defense system in the newly emerged honey bee spring population (short-lived).

The healthy Fb (spring population) honey bees exhibited eight significant correlations among defense genes, where specific interactions accounted for 38%. They included negative correlations with ROS enzymes (similar to Nb) and the involvement of *spz* (Toll pathway) and *defensin-2* (AMP). Compared to Nb, Fb showed reduced gene correlations and specific cross-talk between ROS enzymes, the Toll pathway, and AMP, indicating their immune response during foraging in the short-lived honey bee spring population (Figure 6F).

## 4. Discussion

This study explored the defense systems of long-living (Owb, overwintered honey bees, winter population) and short-living bees (Nb, newly emerged honey bees, and Fb, forager honey bees, spring population) free from *DWV* and *SBV* viruses. We quantified and correlated ten selected pathway reporter genes, including *spz* and *dorsal-1* (Toll pathway), *PGRP-LC* and *relish* (Imd pathway), *domeless* (JAK/STAT pathway), *defensin-2* and *apid-1* (antimicrobial peptides, AMP), and *SOD*, *SOD2*, and *Trxr1* (antioxidative defense genes encoding ROS enzymes), that have been named by molecular profiling [20,21]. The highly expressed genes are of interest for marker selection, which can be achieved through variation, as they tend to exhibit reduced variation between individuals [18,22,23]. In our research, *spz*, *apid-1*, *domeless*, *SOD*, and *relish* genes had the lowest variations; therefore, we suspect their improved gene response and subsequent protein synthesis [24,25,26]. Moreover, the first three genes showed significant differences between Owb, Nb, and Fb honey bees, suggesting their potential as health markers.

In line with previous studies [27,28], *spz* (Toll pathway), and in our research, cytokine receptor *domeless* (JAK/STAT pathway) transcripts were significantly downregulated in Owb compared to the high expression observed in Nb and Fb honey bees. This highlights the potential immune-enhancing effect of pollen nutrition [27] in the spring population of honey bees. The next gene, *apid-1*, an antimicrobial peptide (AMP), exhibited consistent expression between Owb and Nb, but demonstrated significant upregulation in Fb honey bees. Interestingly, our results corroborate a study on 10-day winter and summer honey bees exposed to immune stimuli, where *apid-1* gene expression remained uniform [4].

Despite no significant differences between Owb, Nb, and Fb honey bees, the expression of the *SOD* and *defensin-2* genes was the highest, proving the first role of these enzymes in defense against oxygen free radicals in honey bees [11,29]. Withal, Barroso-Alevado et al. recommended low *defensin-1* and high *relish* gene expressions to identify pre-collapse honey bee colonies [25]. Compromised *defensin-1* and *defensin-2* hamper honey bees’ immune response to pathogens by disrupting the integrity and permeability of pathogenic organisms’ cytoplasmic membranes [30]. In our research, the average expression level of *defensin-2* was higher than the *relish* gene additionally sustaining the health condition of honey bee colonies in our experiment. Upregulated *SOD* and *defensin-2* genes can potentially be used as general health markers without significant differences in season and age.

Previous research noted reduced immune gene expression in the winter population of honey bees [5]. This immune response mechanism in Owb was characterized by the simultaneous downregulation of recognition step gene expression of *spz* and *domeless* in the Toll and JAK/STAT pathways, respectively, along with a wide array of interactions among innate immune genes that totaled 13 correlations. The recognition steps of the JAK-STAT, Toll, and Imd pathways were downregulated by increased expression of the *defensin-2* gene (AMP), which is secreted in the fat body, protects the health of honey bees from pathogens [30], and saves immunity from the overexpression of genes. Our experiments support the findings of Fortezza et al., who observed the strongest activation of JAK/STAT was due to stress [31], not to health conditions. So, high *defensin-2* gene expression can play a role in suppressing innate immunity response in healthy Owb honey bees, opposite to the *apid-1* gene (AMP). The increasing expression of the *apid-1* gene upregulated the expression of the *spz* gene from the Toll pathway recognition step; after that, high *dorsal-1* started to upregulate the *SOD2* gene (ROS enzyme). Both AMP genes control 61% (8 out of 13) of the described defense system correlations, but in opposite directions. So, the reduced defense gene expression in Owb honey bees was compensated for by numerous interactions between innate immunity genes.

Contrary to honey bee winter populations, we hypothesized high defense gene correlations in the spring population of honey bees. However, the number of correlated genes in spring honey bees did not consistently surpass those in winter honey bees. Fb spring populations showed 8 correlations compared to 18 in Nb and 13 in Owb honey bees. The highest characteristics of the defense system were found in Nb honey bees, both in terms of gene expression and correlations. It was provided by the simultaneous upregulation of recognition step gene expression of *spz* and *domeless* in the Toll and JAK/STAT pathways, along with nine significant correlations including ROS enzyme genes and another nine interactions between innate immunity genes. This appears to be a saving mechanism in Nb, when high *SOD*, *SOD2*, and *Trxr1* (ROS enzyme) gene expression downregulated the *domeless* (JAK/STAT pathway) and *apid-1* (AMP) genes, which further downregulated the *spz* (Toll pathway) and *PGRP-LC* (Imd pathway) genes. Presumably, this defense mechanism allows just-emerged individuals to adapt to the beehive microbiome [32].

The few correlations of defense system genes were detected in Fb compared to Nb bees, simultaneously with high gene expression in the recognition step of *spz* (Toll pathway) and *domeless* (JAK/Stat pathway) genes, as well as the *apid-1* (AMP) gene. The same defense system-saving mechanism was found in Fb as in Nb, when high ROS enzyme gene expressions downregulated the innate immunity genes and AMP genes, but the responsible pathways were different. In Fb, high expression of *SOD* and *Trxr1* genes downregulated *spz* (Toll pathway) and *defensin-2* (AMP), which further downregulated *relish* (Imd pathway), *domeless* (JAK/STAT pathway), and *apid-1* (AMD). Notably, downregulation of *defensin-2* and *apid-1* (AMP) gene expression can lead to pathogen intervention and death [25] to promote short-lived mechanisms in spring honey bees. Our results, especially the nine correlations in Nb compared to three correlations in Fb honey bees between ROS enzyme genes with innate immunity genes, support the findings of Guo et al. that the regulatory function of the Imd and Toll pathways for ROS production is crucial in Nb honey bees [33], but not in Fb honey bees (Figure 6C–F). Overall, it additionally revealed the task-specificity of this pattern for Nb and Fb honey bees.

Remarkably, the science terminology has no general terms to categorize long-lived honey bees based on their diapause period and activity phases. The term “winter bee population” was used for honey bees sampled in August in the Czech Republic (Dostalkova et al. 2021), while “winter worker bees” was the term used for honey bees sampled in March in the Republic of Serbia (Orčić et al. 2017). Obviously, the long-lived honey bees sampled in August and March have different physiologies and diapause durations depending on region, but special terms for long-lived honey bees did not clarify this. Moreover, winter timing is a crucial factor for honey bee survival. In the near future, it is essential to standardize the terminology for long-lived overwintered honey bees, similar to how short-lived honey bees were categorized: newly emerged, nurses, and foragers, which will simplify the comparison of scientific data.

## 5. Conclusions

Molecular profiling of the defense system defined (a) molecular markers related to the season and tasks of honey bees, (b) the correlation of pathway performance between long-lived and short-lived honey bees, and (c) the immune strategy of healthy overwintered honey bees. We selected five molecular markers of honey bee health. Of these, two seasonal molecular markers, namely the *domeless* and *spz* genes, exhibited significant downregulation in Owb compared to upregulation in Nb and Fb honey bees. One task-related marker gene, *apid-1*, was recommended because it was downregulated in Owb and Nb but upregulated in Fb honey bees. Two recommended general health markers, *SOD* and *defensin-2*, were upregulated in honey bees to save the first protection line from pathogens. Moreover, in short-lived honey bees, the highest eighteen cross-talking correlations were detected in Nb, which was reduced to eight in Fb with age, and in long-lived honey bees, it decreased to fourteen in Owb. Lastly, overwintered honey bees exhibited suppressed Toll and JAK/STAT pathways (*spz* and *domeless*) and a correlation deficiency between ROS enzyme and innate immunity genes, which characterized the molecular mechanism of longevity in the winter population of honey bees. The five markers require further testing across various honey bee subspecies in different climatic regions to diagnose bee health. It is crucial that honey bee sampling is conducted without colony intervention, especially during low-temperature months like winter. Beekeepers can use this information to make timely adjustments in nutrients or heating to prevent seasonal losses. Additionally, the molecular markers of health in overwintered honey bees revealed signs of longevity, continuing the anti-aging trend in both science and practice.

## Figures and Tables

**Figure 1 biomolecules-14-00019-f001:**
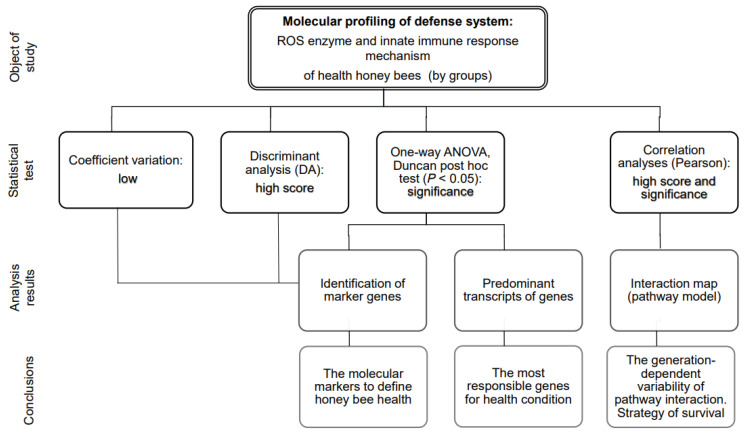
The statistical tests and criteria used for molecular profiling analysis and conclusions.

**Figure 2 biomolecules-14-00019-f002:**
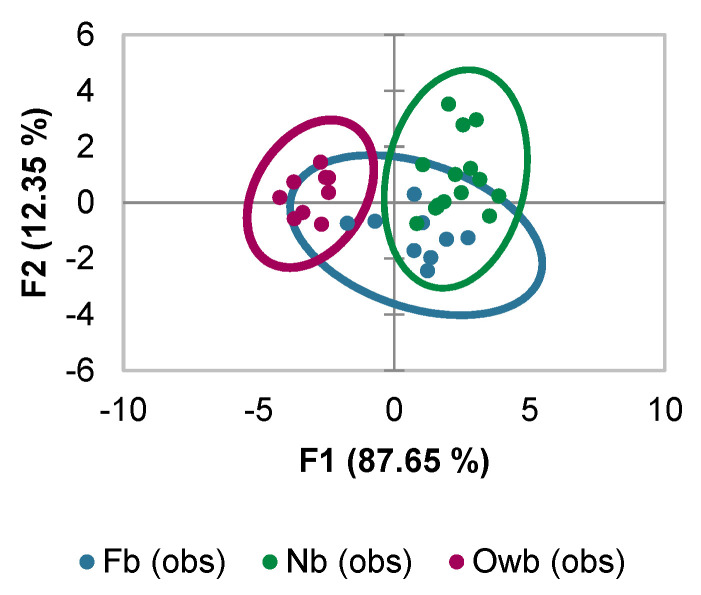
Discriminant Analysis (DA) of overwintered (Owb), newly emerged (Nb), and forager (Fb) honey bees using the expression of ten defense system genes: *spz*, *dorsal*−*1*, *PGRP-LC*, *relish*, *domeless*, *defensin*−*2*, *apid*−*1*, *SOD*, *SOD2*, and *Trxr1*.

**Figure 3 biomolecules-14-00019-f003:**
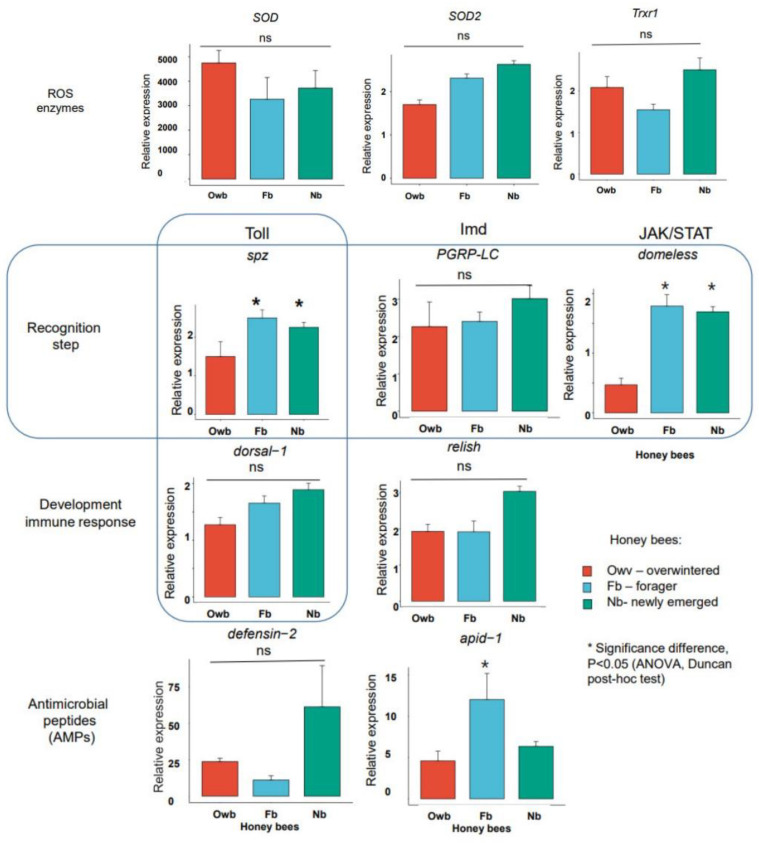
Bar plots for defense system gene expressions via means and standard deviations. Ns—not significant, Nb—newly emerged honey bee spring population, Fb—forager honey bee spring population, Owb—overwintered honey bee winter population. * significant differences (*p* < 0.05), ns—not significant differences (*p >* 0.05).

**Figure 4 biomolecules-14-00019-f004:**
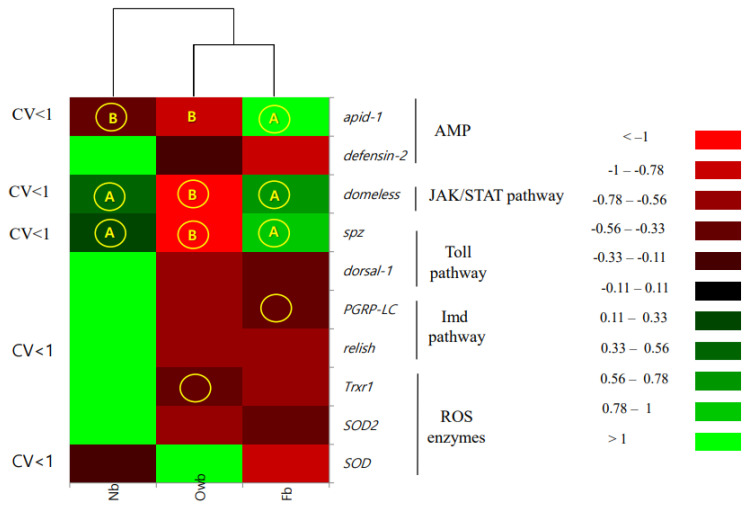
Heat map of gene expressions in the winter (Owb) and spring honey bee populations (newly emerged—Nb and forager—Fb). Gene names and pathways are indicated on the right. Boxes marked with (**A**) represent statistically significant upregulation, and those marked with (**B**) represent downregulation due to generation or age, with a *p*-value less than 0.05. Genes marked with (O, yellow color) were selected by using Discriminant Analysis (DA). Genes with CV < 1 indicate a coefficient of variation lower than 1, observed in all honey bee groups for gene expression.

**Figure 5 biomolecules-14-00019-f005:**
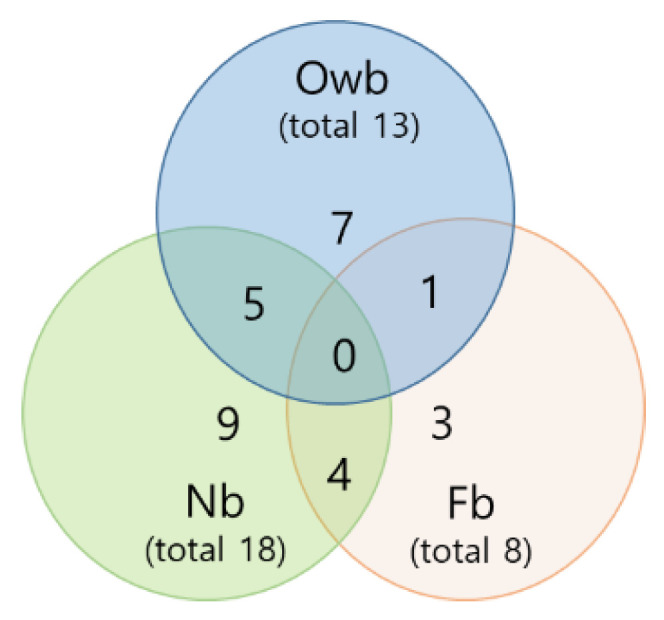
Venn diagram comparing the honey bee molecular profiling based on interactions between ten genes of the defense system through significant correlations (moderate and higher). Owb—overwintered bees of the winter population; Nb—newly emerged bees; and Fb—forager bees of the spring population.

**Figure 6 biomolecules-14-00019-f006:**
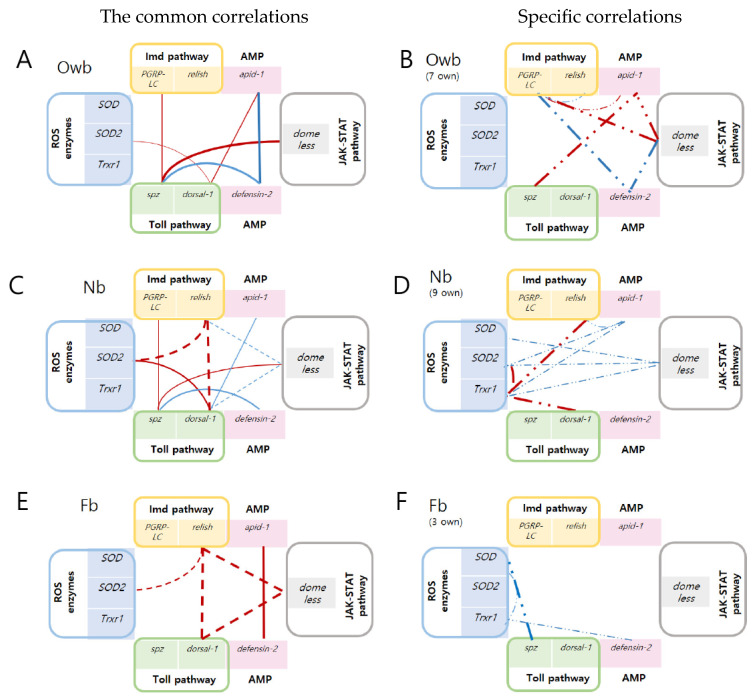
Correlation analysis of the honey bee defense system through reporter gene interactions. (**A**,**B**) represent the winter population (Owb); (**C**,**D**) represent the spring population of newly emerged bees (Nb); and (**E**,**F**) represent the spring population of forager (Fb) honey bees. The solid line indicates common interactions of Nb and Fb with the Owb correlations. The dashed line represents common interactions within the spring population (Nb and Fb). Negative correlations are shown in blue, and positive correlations in red. Low and negligible correlations were excluded.

**Table 1 biomolecules-14-00019-t001:** PCR primer sequences.

	Locus	5′-3′	Primers	Amplicon Length, bp	NCBI Reference Sequence
1	*defensin-2*	F	ACCGCTGCTACCACTACGACA	139	NM_001011638.1
2		R	GCCATTTCTGCAACTACCGCCT		
3	*relish*	F	TCCATTGCATGCAGCACTTCG	264	XM_026444175.1
4		R	ACACATGCACCAGCTTCAGGA		
5	*dorsal-1*	F	TGCAGCAAGTGGAACAACCAGT	114	XM_006566999.3
6		R	CAGGCCTACCTGCACCGAGA		
7	*domeless*	F	GCCGCTGCTCTTTGGCATCT	238	XM_006567690.3
8		R	GCCAAATTGTTGTTCCAACAGCCC		
9	*apid-1*	F	TTGTTGTTACCTTTGTAGTCGCGGT	70	NM_001011642.1
10		R	AGGCGCGTAGGTCGAGTAGG		
11	*PGRP-LC*	F	TGCAATGCGATGGCGACACA	105	XM_026441962.1
12		R	AGCGACTTGAGCACACCACAC		
13	*spz (spaetzle)*	F	TGGACGACAGCCCTCTTTGTCA	371	XM_006565534.3
14		R	GCGCCTTCGACGTGACGATT		
15	*SBV*	F	GTGGAACCCGAGTGTTTTGTAACCC	156	KY273489.1
16		R	AAGCTAAAAGCGTCCACTCTGTACTCT		
17	*DWV*	F	TGT GAA GTG GCG GAC GTT ACA GA	211	KT215904.1
18		R	GTA TTC TGG ACC CCA TCC GAA TGC		
19	*β-actin*	F	GGATTCCTATGTTGGTGATGAAGCCC	177	NM_001185145.1
20		R	GGTGCCTCAGTAAGAAGTACCGGATG		
21	*SOD*	F	GCAGTGTGCGTTCTTCAGGGT	86	NM_001178027.1
22		R	TGACCGGTGACCTTCACGGA		
23	*SOD2*	F	GGCGGTAAACCAGACGCTGC	126	NM_001178048.2
24		R	TCCAAGCCAACCCCAACCAGA		
25	*Trxr-1*	F	CCTGTTGCTATACATGCGGGTCG	141	XM_006563201.3
26		R	TGCTGCTTCTTCGCTAAGGCCA		

**Table 2 biomolecules-14-00019-t002:** Coefficient of variation (CV) gene expression (marker candidates bolded).

Name of Gene/Honey Bee Group	*SOD*	*SOD2*	*Trxr1*	*spz*	*dorsal*−*1*	*defensin*−*2*	*domeless*	*apid*−*1*	*PGRP-LC*	*relish*
Owb	**0.304**	0.175	0.353	**0.722**	0.286	0.269	**0.671**	**0.821**	0.286	**0.717**
Nb	**0.720**	1.713	1.289	**0.216**	1.770	1.716	**0.192**	**0.436**	1.711	**0.342**
Fb	**0.771**	0.116	0.241	**0.235**	0.214	0.741	**0.306**	**0.293**	0.437	**0.738**

## Data Availability

The data can be made available upon reasonable request to the corresponding author.

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
