# Peer review of "Monitoring Immune Modulation in Season Population: Identifying Effects and Markers Related to Apis mellifera ligustica Honey Bee Health"

_biomolecules, 2023, doi:10.3390/biom14010019_

Round 1
Reviewer 1 Report
Comments and Suggestions for Authors
In my opinion, the manuscript covers an extremely interesting and current topic. The test is representative, the sample size is adequate. The authors performed their tests with 12 colonies. The description of the RNA and cDNA tests is understandable and easy to follow. The statistical analysis of the results is adequate. The discussion section is detailed and comprehensive.
Critical comments:
In my opinion, the quality of Figure 1 is not adequate, the shaded text boxes are misleading, I recommend editing the figure. The captions in Figure 3 are illegible, so they are not informative. The entire diagram system requires revision, it cannot be followed in this form.
The manuscript contains some typos.
Author Response
Dear Reviewer 1!
Thank you for the time and effort spend reviewing our manuscript number biomolecules-2743289 and suggesting some important points to consider.
Please, our answers on comments you can find in the attached file.
We look forward to hearing from you.
Sincerely,
Professor Hyung Wook Kwon

Reviewer 2 Report
Comments and Suggestions for Authors
Monitoring Immune Modulation in Season Population: Identifying Effects and Markers on Apis mellifera ligustica Honey Bee Health
Good and original idea. Modern molecular genetic analyses were used and great results were obtained. The data are well processed, and the results have potential not only from a scientific point of view but also in beekeeping practice.
However, the paper is not carefully written (the way of presenting is inappropriate, the whole manuscript suffers from multiple and serious syntax errors; even the abstract and conclusion are vaguely written). Besides, I would ask authors to check the meaning of Molecular Profiling“ and reconsider the adequacy of usage that phrase in this manuscript (13 times repeated).
- Lines 21-22 („to protect them from pathogens“ should be deleted);
- Lines 242 - Here I found the most important sentence (five molecular markers of honey bee health) that I suggest to repeat both in Abstract and the Conclusion);
- Line 245 - „to protect them from pathogens“ should be deleted;
- Table 1: In the last column „Target“ should be replaced with „NCBI Reference Sequence“.
- Line 521: Reference is inproperly written: ’Review’ is a part of article title (should not be written italic). The journal name is missing. The volume and pages are wrong. Please replace
’Review 2019, 37, 521–33’ with ’Revista Mexicana De Ciencias Pecuarias 2019, 10, 705-728’. Please see the original PDF:
- References are not uniformly written (e.g. names of journals are written in full in some references and in others - abbreviations are written; pagination is not uniformly written).
SYNTAX/LINGUISTIC ERRORS SHOULD BE CORRECTED IN FOLLOWING PARTS:
Line 13-16;
Lines 33-36;
Lines 47-48;
Lines 98-99;
Line 101;
Lines 254-255;
Lines 264-269;
Lines 275-276;
Line 281;
Lines 294-301;
Line 304;
Line 314;
Lines 320-326;
Lines 351-352;
Line 356;
Line 361;
Lines 377-378;
Line 380;
Lines 384-386;
Lines 391-396;
Lines 420-428;
Lines 433-434;
Title of Figure 2;
Title of Figure 3;
Title of Figure 4;
Title of Figure 5;
Title of Table 2.
Comments on the Quality of English LanguageТhe whole manuscript suffers from multiple and serious syntax errors.
SYNTAX/LINGUISTIC ERRORS SHOULD BE CORRECTED IN FOLLOWING PARTS:
Line 13-16;
Lines 33-36;
Lines 47-48;
Lines 98-99;
Line 101;
Lines 254-255;
Lines 264-269;
Lines 275-276;
Line 281;
Lines 294-301;
Line 304;
Line 314;
Lines 320-326;
Lines 351-352;
Line 356;
Line 361;
Lines 377-378;
Line 380;
Lines 384-386;
Lines 391-396;
Lines 420-428;
Lines 433-434;
Title of Figure 2;
Title of Figure 3;
Title of Figure 4;
Title of Figure 5;
Title of Table 2.
Author Response
Dear Reviewer 2!
Thank you for the time and effort spend reviewing our manuscript number biomolecules-2743289 and suggesting some important points to consider.
Please, our answers on comments you can find in the attached file.
We look forward to hearing from you.
Sincerely,
Professor Hyung Wook Kwon
